# Interactions with Microbial Proteins Driving the Antibacterial Activity of Flavonoids

**DOI:** 10.3390/pharmaceutics13050660

**Published:** 2021-05-05

**Authors:** Giuliana Donadio, Francesca Mensitieri, Valentina Santoro, Valentina Parisi, Maria Laura Bellone, Nunziatina De Tommasi, Viviana Izzo, Fabrizio Dal Piaz

**Affiliations:** 1Department of Pharmacy, University of Salerno, 84084 Fisciano, Italy; gdonadio@unisa.it (G.D.); vsantoro@unisa.it (V.S.); vparisi@unisa.it (V.P.); mbellone@unisa.it (M.L.B.); detommasi@unisa.it (N.D.T.); 2Department of Medicine and Surgery, University of Salerno, 84082 Baronissi, Italy; fmensitieri@unisa.it (F.M.); vizzo@unisa.it (V.I.); 3PhD Program in Drug Discovery and Development, Department of Pharmacy, University of Salerno, 84084 Fisciano, Italy

**Keywords:** flavonoids, antibacterial activity, bioactive natural compounds, enzyme inhibitor, efflux pumps, ATP synthetase, DNA gyrase, antibiofilm activity

## Abstract

Flavonoids are among the most abundant natural bioactive compounds produced by plants. Many different activities have been reported for these secondary metabolites against numerous cells and systems. One of the most interesting is certainly the antimicrobial, which is stimulated through various molecular mechanisms. In fact, flavonoids are effective both in directly damaging the envelope of Gram-negative and Gram-positive bacteria but also by acting toward specific molecular targets essential for the survival of these microorganisms. The purpose of this paper is to present an overview of the most interesting results obtained in the research focused on the study of the interactions between flavonoids and bacterial proteins. Despite the great structural heterogeneity of these plant metabolites, it is interesting to observe that many flavonoids affect the same cellular pathways. Furthermore, it is evident that some of these compounds interact with more than one target, producing multiple effects. Taken together, the reported data demonstrate the great potential of flavonoids in developing innovative systems, which can help address the increasingly serious problem of antibiotic resistance.

## 1. Introduction

Natural flavonoids are secondary metabolites widely produced by plants. These are polyphenolic compounds generally characterized by a three-cyclic structure, consisting of two phenyl/benzyl rings (A and B) connected by a heterocycle (C). Depending on the level of unsaturation and oxidation, and on the position of the different substituents, flavonoids can be grouped into various subclasses (Figure 1): flavones (i.e., apigenin, luteolin, bacalein, tangeritin, diosmetin), in which the B ring is linked to the C ring at position C2; isoflavones (i.e., genistein, daidzein), whose B ring is linked to the C3 position of C ring; flavonols (i.e., myricetin, kaempferol, quercetin) that share a 3-hydroxyflavones backbone; flavanols, which are also named catechins or flavan-3-ols (i.e., epicatechin, epicatechin-3-*O*-gallate or ECG, epigallocatechin-3-*O*-gallate or EGCG) and are derived from hydroxylation at C3 of flavans, benzopyran derivatives sharing the 2-phenyl-3,4-dihydro-2H-chromene skeleton; flavanones (i.e., naringenin, hesperetin, sakuranetin, pinocembrin), similar to flavones, lacking C-C double bonds on the C ring; flavanonols (i.e., dihydrokaempferol, dihydroquercetin or taxifolin, silymarin), which are flavanones bearing a hydroxylic group on the C3; anthocyanidins (i.e., cyanidin, delphinidin, pelargonidin, europinidin), which share the flavanols structure, bearing an oxonium ion on oxygen in the C ring; lastly, chalcones and dihydrochalcones (i.e., phloretin, licochalcone A, B, C, D, butein, bartericin A) derive from the structure of anthocyanidins at higher pH, at which the C ring opens, and an aromatic ketone is formed.

Flavonoids are possibly the most abundant secondary metabolites in plants, and their synthesis is finely regulated, varying with environmental conditions. These compounds display a range of different physiological roles in vivo, being generlly involved in different types of abiotic (visible and ultraviolet light, heat) or biotic (pathogens) stress-induced responses [1,2]. Some flavonoids are produced in a species-specific mode (flavanones in citrus fruits, isoflavons in soya, phlorizin in apples); others, such as naringin and quercetin, are ubiquitous in most plants. The huge structural diversity and wide biological activity of flavonoids becomes evident if all possible backbone modifications are considered. Moreover, different functional groups are often bound to the polyphenolic structure to fine-tune their reactivity, stability, solubility, and bioavailability. Glycosylated and methylated flavonoids are often produced by the different plant families [3,4].

A role of primary importance for these compounds in plant cells is the defense against reactive oxygen species (ROS). The abundance in hydroxyl groups and conjugated double bonds in their chemical structure make flavonoids effective in ROS scavenging, metal chelation, and quenching of lipid peroxidation [5,6]. Another prominent function of flavonoids, correlated with the previous one, is the inhibition of auxin transport in plant organs. Auxin co-regulates growth and light-orientation in plants, and different studies have highlighted that the interplay between this hormone and flavonoids, whose production are both stimulated by UV light, contributes to the regulation of the UV-induced morphogenesis [7,8]. Among flavonoids, anthocyanins constitute the major constituent of plant pigments, along with chlorophyll and carotenoids. The differential enrichment in pelargonidin, cyanin, or delphinidin-like anthocyanins determines flower color, playing a central role in pollinator attraction and plant reproduction [9]. Physiological roles of flavonoids are not limited to plant aerial parts; they are produced also in the roots, where they function as chemo-attractors for symbionts recruitment from soil, such as for rhizobia in leguminous plants [10]. Finally, flavonoids are endowed with diverse antimicrobial properties, which have been extensively verified against a wide range of Gram-positive and Gram-negative pathogens and some viruses [11]. The variety of substituents that could be attached to the polyphenolic backbone regulates flavonoids hydrophilicity and modifies their biological and steric properties, varying the range of target ligands [12]. In this context, also flavonoids antibacterial activity involves different pathways, depending on the peculiar chemical features of each compound. More hydrophobic flavonoids preferentially interact with phospholipidic bilayers, sometimes disrupting membrane integrity [13]. Conversely, an increasing number of hydroxyl groups on the phenol rings are related to an enhanced reactivity toward proteins, enzyme-inactivation effects, or receptor shielding [14].

Natural flavonoids have gained increasing recognition in recent years for their health-related activities, such as their potential antioxidant, antitumor, and anti-inflammatory, antimicrobial, and antifungal properties [15,16,17,18,19,20]. Several reports describe flavonoid potential to be used as drugs and nutraceutical compounds. Therefore, flavonoids, along with coumarins, terpenoids, and phenolic acids, are among the main constituents of plant extracts extensively used in folk medicine [21]. Some examples encompass Ayurveda, traditional Chinese medicines (TCM), African traditional medicine, and Brazilian traditional medicine [22]. However, flavonoids’ application to the clinical research is still limited. Different reasons lie for this scarce application. First, in most studies, the mechanisms of action of tested flavonoids were not fully elucidated to a molecular level. Moreover, even though different reports highlight flavonoids to be promising therapeutic agents due to their lack of toxicity [23,24], this is probably also due to their scarce bioavailability in the organism, leading to a rapid metabolism and excretion [25]. Flavonoids are produced in a glycosylated form in plants, to increase their hydrophilicity and solubility [26]. In humans, endogenous β-glycosidases and α-l-arabinosidases in the small intestine are responsible for removing glucose (or possibly arabinose or xylose) moiety from flavonoids to allow their effective absorption [27]. However, these enzymes are not able to cleave other terminal sugar units, such as rhamnose, which is often present in plant flavonoids as single residues or as rutinose disaccharidic units [28]. Therefore, the bioavailability of rhamnosylated flavonoids is strongly limited, and they arrive unabsorbed in the colon, where they are converted to more bioactive forms only by the microflora [27]. Moreover, the presence of the sugar moiety that subtracts hydroxyl residues from the catecholic ring, other than being beneficial for their absorption, also hampers flavonoids antioxidant and anti-inflammatory qualities, reducing their interaction with possible molecular targets [29].

Among all the therapeutic activities of plant flavonoids, their antimicrobial and antibacterial action became of particular interest in latest years, especially considering the concerns toward an increasing antibiotic resistance phenomenon.

Gram-positive bacteria are responsible for post-operative wound infections, toxic shock syndrome, endocarditis, osteomyelitis, and food poisoning [30]. Gram-negative bacteria cause infections including pneumonia, bloodstream infections, wound or surgical site infections, and meningitis in healthcare settings; they are resistant to multiple drugs and are increasingly resistant to most available antibiotics [31,32]. The discovery of antibiotics has been one of the foremost medical advances of the 20th century, and different antibiotics exercise their inhibitory activity on different pathogenic organisms. There are different antibiotic mechanisms that involve specific target molecules, most of which are implicated in the inhibition of cell wall synthesis, protein synthesis, fatty acid synthesis, ribonucleic acid synthesis, carbohydrate transport, sterol biosynthetic pathway, and intermediary metabolism. However, in latest years, the effectiveness of antibiotic treatment is being constantly jeopardized by the increase of multidrug resistance mechanisms in bacteria, also due to indiscriminate use of commercially available antimicrobial drugs. Examples are the growing prevalence in the environment of multidrug resistant and nosocomial strains, such as MRSA (Methicillin-resistant *Staphylococcus aureus*), which has acquired a multiple drug resistance to a variety of β-lactam antibiotics, MDR (Multidrug-Resistant) *Pseudomonas aeruginosa*, DR (Drug-Resistant) *Streptococcus pneumoniae*, and different MDR uropathogenic *E. coli* strains, which have developed parallel resistance to different classes of antibiotics [33]. In addition, biofilm-producing strains are highly tolerant to antibiotics. The prolonged treatment of resistant strains results in a health risk for patients and in an additional premise for the further development of antibiotic resistance in microbes [34]. Despite the efforts of the current research for novel antibiotics, none of these new drugs has been clinically approved in the last three decades. Therefore, there is a growing interest in evaluating a combined use of antibiotics and natural compounds, which could enhance the antimicrobial activity overcoming drug resistance [35]. In this framework, natural bioactive compounds and plant secondary metabolites have always represented a powerful reservoir in drug discovery [11].

Natural polyphenols, and especially plant flavonoids, are synthesized also in response to antimicrobial infections and have been shown to possess antibacterial activity against a broad range of microorganisms [36,37,38,39]. Moreover, some reports describe that even flavonoids that do not exert a significant antimicrobial activity on their own, when used in combination with an antibiotic, are able to potentiate its effect and, in some cases, to revert bacterial resistance toward the specific antibiotic [40]. Interestingly, in normal conditions, this kind of synergistic action mechanism offers a great advantage, since it is unlikely to pose a selective pressure for the development of further resistance in bacteria. However, the specific mechanism of action of many phytochemicals and flavonoids is sometimes still unclear. In this framework, in the last few years, several research studies have been conducted in different countries to prove such antimicrobial efficiency [39,41].

This review is focused on describing the interactions between flavonoids and proteins depicted so far, which are involved in the antimicrobial activity of these secondary metabolites. Different studies highlighted the specific interactions of flavonoids with prokaryotic proteins involved in mechanisms of growth and pathogenesis. In general, flavonoids protein interactome encompass three main groups of proteins and enzymes central in bacterial growth and metabolism: enzymes involved in DNA and protein metabolism (i.e., bacterial topoisomerases, helicases, DNA gyrases, and ribosomes); membrane proteins and enzymes involved at various extent in cellular transport, bioenergetic, homeostasis maintenance, and cell wall and lipid metabolism (i.e., efflux pumps and transporters, ATP synthase, cytochrome c, β-ketoacyl-acyl carrier protein synthases); other targets involve beneficial effects against microbial pathogenesis, such as their action against toxin and biofilm production.

In this review, the information present in the scientific literature has been divided based on different pathways perturbed by the action of flavonoids on cells and bacterial systems (Figure 2).

All flavonoids discussed in this review and the respective molecular targets are reported in Appendix A.

## 2. Bacterial Membrane

The bacterial plasma membrane is an overly complex microenvironment. Other than serving as a flexible barrier from the external environment, and being a key regulator in cell osmoregulation and transport, it also sustains cell energy generation, lipid biosynthesis, and cell wall maintenance and metabolism. Moreover, all the cellular external molecular trafficking, even if not here regulated, ultimately takes place in the plasma membrane. It includes all the exocytosis mechanisms involved in toxins secretion, quorum sensing, and biofilm formation. For these reasons, it is worth discussing this cellular compartment first.

Perturbations in membrane integrity and membrane enzymes activities pose significant threats to cellular metabolism, ultimately leading to cell death. A consistent group of antibiotics (penicillins, cephalosporins, polymixins, and antimicrobial peptides) has the bacterial membrane and cell wall as primary targets. In addition, the antimicrobial activity of flavonoids was supposed to rely upon their ability to perturb bacterial homeostasis and membrane permeability toward drugs and antibiotics.

The interactions of flavonoids with the bacterial membrane proteome have not yet been fully understood; however, three principal mechanisms could be outlined, as described in the following paragraphs:-Interaction and inactivation of bacterial efflux pumps transporters, in both Gram-negative and Gram-positive bacteria.-Physical membrane disruption and depolarization caused by a direct interaction with lipidic bilayers, resulting in a change in membrane fluidity, or by a perturbation in cell envelope metabolic pathway, through the inhibition of fatty acids synthesis.-Inhibition of ATP-synthase and disruption of the bioenergetic status, by energy transduction mechanisms uncoupling.

### 2.1. Efflux Pumps and Flavonoids Interaction

Efflux pumps (EP) are single- or multicomponent channel proteins present in both Gram-positive and Gram-negative bacteria. They actively modulate drugs and solutes efflux from the cell, thus allowing bacteria to regulate their internal microenvironment, to maintain membrane potential, and to extrude toxic compounds and signal molecules [42]. The acquisition over time of a growing subset of different EP is one of the main mechanisms of antibiotic resistance outbreak in bacteria. These transporters constitute tools to avoid antibiotic intracellular accumulation [43]. Therefore, the efflux pumps’ inhibition properties reported for some flavonoids often allows potentiating antibiotic activity and, in some cases, to obtain a full recovery of the bacteria susceptibility to antibiotic drugs.

Soto deeply reviewed the six families of EP drug/proton antiporters present in bacteria [42]. The principal ones encompass the ATP-binding cassette (ABC) superfamily, the resistance–nodulation–division (RND) superfamily, which is only present in Gram-negative bacteria, and the major facilitator superfamily (MFS) such as Nor A pump, which is only present in Gram-positive bacteria.

Despite their advantages as antibiotic-potentiating agents, not many efflux pumps inhibitors were identified and are currently used in therapy. Some examples are Phe-Arg β-naphthylamide (PAβN) (which inhibits resistance–nodulation–division (RND) superfamily EP), reserpine (an ATP-dependent pumps inhibitor), and verapamil (P-glycoprotein inhibitor) [44,45]. These molecules act with different mechanisms, but all of them lead to a final accumulation increase and extrusion decrease of EP substrates [42]. Moreover, some efflux pumps inhibitors directly cause also a decrease in biofilm formation and a disruption of quorum-sensing mechanisms in several bacteria (i.e., *Escherichia coli*, *Klebsiella pneumonia*, *Staphylococcus aureus*, *Pseudomonas aeruginosa,* and *Pseudomonas putida*). This evidence clearly suggests that several bacterial metabolic pathways ultimately co-localize and are co-regulated at the plasma membrane level [27,46,47,48].

In the last decade, different papers highlighted the efflux pump inhibitory activity of flavonoids isolated from a variety of natural sources. Many reports are focused on Gram-positive bacteria, in particular on *S. aureus* and methicillin-resistant *S. aureus*. Studies of the effect of flavonoids as EP inhibitors were carried out also in *Mycobacterium tuberculosis* and related strains. The mycobacterial cell wall has a peculiar structure, and it is a harsh pathogen that is highly resistant to different antibiotics; the intrinsic and acquired resistance of this pathogen is partly due to the great variety of multidrug efflux pumps present in this genus [49]. Conversely, only few studies were carried out on the EP inhibitor activity of flavonoids in Gram-negative bacteria such as *E. coli*, *Proteus mirabilis*, *P. aeruginosa,* and *Salmonella enterica* [40,50].

Flavonoid inhibition on EP is by far the most documented in *S. aureus*. Brown et al. and Lechner et al. described the efflux pump inhibitory activity of hydroxylated and methoxylated flavonoids isolated from different plant sources [51,52,53]. Data highlighted the following ranking in EP efficiency: myricetin > rhamnetin > kaempferol > apigenin, with luteolin and quercetin having a negligible activity. However, the authors outlined that the individual flavonoids could only be partially responsible for the overall activity observed, which is much higher in the phytocomplex. In fact, in these latter, an EP inhibition at concentrations below the MIC (minimum inhibitory concentration) value was observed, thus suggesting the occurrence of some synergistic interaction between the different compounds [52]. Another study described the role as modulators of drug resistance in *S. aureus* of some methoxylated flavonoids (apigenin, genkwanin, 7,4′-dimethylapigenin, trimethylapigenin, cirsimaritin, tetramethylscutellarein) extracted from *Praxelis clematidea* (Griseb.) R.M.King & H.Rob (Compositae) [53]. Isolated methoxylated flavones showed a lack of antibacterial activity against the tested strain but, when added to the growth medium of bacteria over-expressing the *norA* gene encoding the NorA efflux protein, a reduction in the MIC of at least 2-fold (and up to 16-fold) was observed for norfloxacin and ethidium bromide. This evidence confirmed the action of the investigated compounds as EP inhibitors, with tetramethylscutellarein being the most active compound, followed by cirsimaritin, trimethylapigenin, and 7,4′-dimethylapigenin [53].

Taken together, these results suggested that the EP inhibition action of flavonoids seems to be related to their substituents pattern. More specifically, a large number of -OH substituents in the B-ring, as in myricetin, enhanced the efflux-inhibiting efficiency in *S. aureus*. Moreover, methoxylated flavonoids apparently modulated EP activity even more efficiently, as demonstrated for tetramethylscutellarein. The authors highlighted that this increased efficiency could be due to the higher lipophilicity of this molecule, due to the presence of methoxyl groups. Lipophilicity is in fact a common feature of several efflux pump inhibitors and may be a key factor for EP inhibition in Gram-positive bacteria [54]. The presence of a methoxyl-group at the 4′ position also was crucial for flavonoids activity [55].

Another complex and prenylated flavonoid-artonin I-was found to inhibit the bacterial EP in MDR *S. aureus*. Artonin I was isolated from the leaves of a medicinal plant *Morus mesozygia* Stapf. (Moraceae) [56]. The authors of that research performed the EtBr assay in methicillin-resistant *S. aureus*, using artonin I in combination with antibiotics belonging to different classes (oxacillin, neomycin, erythromycin, ciprofloxacin, gentamicin, sulfamethoxazole, chloramphenicol, and amikacin). The flavonoid significantly increased bacterial sensitivity toward drugs by blocking EP, according with the increased penetration of ethidium bromide (EtBr) into the cells. Therefore, artonin I was able to revert the multidrug resistance in methicillin-resistant *S. aureus*, even at low concentrations, as evident from the 1000-fold decrease in MIC values of some antibiotic drugs, when used in combination with artonin I [56].

It is worth noting that even though a clear EP effect was shown, artonin I displayed also other synergistic actions as a disruptor of membrane potential and inducer of high oxidative stress. In fact, the authors also showed that artonin I increased the intracellular ROS level in a dose-dependent manner and had a loss of membrane potential, which may cause DNA and membrane damage [46]. Loss of envelope integrity was evident from SEM and TEM analyses, showing a highly perforated, porous, and leaky membrane, ultimately leading to cell death [56]. Therefore, this can be considered a case of multiple mechanisms responsible for the antibacterial action of artonin I.

Another flavonoid selectively inhibiting the NorA MDR pump in *S. aureus* was described by Stermitz et al. [57]. 5′-Methoxyhydnocarpin (5′-MHC) was identified as an EP MDR-NorA inhibitor and was able to revert resistance to quinolones and antiseptics. Again, the methoxy substituents and hydrophobic nature of 5′-MHC was important for its EP inhibitor properties, even though the authors suggest that it acts with a different binding mechanism when compared to the known NorA MDR substrates.

A central role for methoxy-substituted flavonoids was also suggested for the action on the MDR efflux pumps in *Mycobacteria* strains. As an example, the inhibitory action of the isoflavone biochanin A in potentiating the antibacterial activities of norfloxacin and berberine was found in *Mycobacterium smegmatis*, whereas luteolin, daidzein, and resveratrol exerted only a limited activity [44]. According to the authors, the para-methoxy group in the B-ring of biochanin A, which is not present in the structure of the other tested compounds, strongly enhanced the EP-modulating activity of this isoflavone. Moreover, Solnier et al. outlined the EP inhibitory activity of diverse methoxylated flavonoids, whose lipophilic nature should exert a higher affinity for the lipid-rich mycobacterial cell envelope: skullcapflavone II, nobiletin, tangeretin, wogonin, and the non-methoxylated baicalein. In this study, the moderately pathogenic strains *M. smegmatis*, *M. aurum*, and *M. bovis* BCG were used as models for *M. tuberculosis*, to analyze efflux-mediated resistance, and a significant EP activity impairment by the investigated compounds was observed. Interestingly, even though the non-methoxylated baicalein exerted the stronger antimicrobial activity showing the lower MIC values, the most efficient EP inhibitors were the highly methoxylated skullcapflavone II and nobiletin. Again, the presence of an additional 3′-methoxy group in the B-ring significantly improved the efflux inhibitory activity of the flavonoids. Based on their results, the authors pointed out that skullcapflavone II and nobiletin flavonoids could be considered antimycobacterial compounds, which is particularly interesting because of the current lack of clinically approved EP inhibitors for this pathogenic bacterium [58].

As previously mentioned, few reports are currently available on flavonoids EP inhibitory action in Gram-negative bacteria. A recent study from Maisuria et al. extensively described the role of cranberry proanthocyanidins (cPAC) in interfering with the antibiotic resistance mechanisms of Gram-negative bacteria (*E. coli*, *Proteus mirabilis*, and *P. aeruginosa*). These bacteria indeed have different intrinsic resistances toward tetracycline, aminoglycosides, β-lactams, quinolones, and polymyxins. In a first paper, the authors demonstrated that cPAC combined with a tetracycline was able to completely prevent the development of resistance in *E. coli* and *P. aeruginosa*; therefore, this therapeutic mixture was also proposed as a natural remedy for urinary tract infections [59]. Subsequently, the same authors described the specific role played by two cPAC, namely the proanthocyanidins PAC A and PAC B purified from the American cranberry fruit *Vaccinium macrocarpon L*. (Ericaceae), in EP impairment of the resistance–nodulation–division (RND) family efflux pumps AcrAB–TolC [60,61] and MexAB–OprM [62]. To this purpose, the authors firstly showed that cPAC strongly enhanced the effectiveness of several antibiotics, providing a 32/64-fold reduction in MIC for sulfamethoxazole, nitrofurantoin, gentamicin, kanamycin, tetracycline, azithromycin, and fosfomycin against *E. coli*, *P. mirabilis,* and *P. aeruginosa*. Moreover, they observed that cPAC efficacy was drug- and strain-dependent (e.g., cPAC potentiated fosfomicin against *P. mirabilis* but not against *P. aeruginosa* or *E. coli*), supporting the hypothesis that the flavonoids acted through a well-defined mechanism. MIC analysis performed on *P. aeruginosa* PA14 efflux pump overexpressing mutants confirmed the correlation between cPAC treatment effects and the efflux pump activity, thus validating this hypothesis. The authors described an in silico docking analysis of the interaction between the A-type dimeric cPAC (PAC A) and the efflux pump protein complexes AcrAB–TolC of *E. coli* and MexAB–OprM of *P. aeruginosa*. To the best of our knowledge, this was the only example of docking study of EP component with flavonoidic compounds. As representative, here, we reported the image of the authors docking analysis of cPAC in the AcrAB–TolC efflux pump in Figure 3. Although such analysis is based on a steric and energetic in silico simulation of substrate interaction with a protein crystal structure, it could provide indicative data on the binding mode between two or more compounds and on the stability of the resulting complexes. Using this approach, the most probable regions of interaction of PAC A with the protein complexes were identified in the exit duct of TolC and OprM, and in the antibiotic/substrate binding site of AcrB and MexB. Interestingly, those regions were the same involved in the binding of the antibiotics whose efficiency was enhanced by PAC A. Conversely, two antibiotics non-potentiated by PAC A in *E. coli* were shown to interact with a distal binding pocket of AcrAB–TolC, and their binding to the protein complex was not affected by the presence of the proanthocyanidin [40]. These evidences further supported the hypothesis that binding of PAC A in the EP binding pocket interferes with the preferred binding position of the potentiated antibiotics, thus leading to an efflux inhibition effect.

### 2.2. Flavonoids Interactions in Cell Envelope Metabolism Disruption

As previously mentioned, flavonoids could exert a disruptive effect on bacterial membranes, acting at different extents and with diverse mechanisms. Therefore, in this paragraph, we will discuss an heterogenous range of flavonoids interactions that ultimately lead to cell envelope damage. Several papers describe a plethora of macroscopic effects, such as membrane permeabilization, membrane rupture, or uncontrolled boost of ROS molecules, with consequent oxidative damage and lipid peroxidation events. However, few of these studies clearly relate these effects to a specific mechanism of action or target interaction. The described effects of flavonoids on membrane can be grouped in (1) membrane permeabilization and rupture; (2) inhibition of membrane and cell wall biosynthetic mechanisms; (3) direct interaction with membrane proteins, or extra and intracellular protein aggregation phenomena; and (4) uncontrolled ROS generation.

Given this variety of effects, parallelly and similarly evaluated in most cases by the authors, we prefer here to review the different papers based on flavonoid classes. It is worth noting that in this case, differently from what we observed in EP inhibitory activity, most flavonoids described for increased membrane disruption effects are hydroxylated or prenylated but not methoxylated.

Fathima et al. and Sinsinwar et al. described the membrane disruption antimicrobial activity of catechin isolated from different plants [63,64]. Catechin is a hydroxylated flavanol known for its potent free radical scavenger activity and antibacterial activity toward Gram-positive and Gram-negative bacteria [65]. Fathima et al. studied its action against *Bacillus subtilis* and *E. coli*, showing some kind of concentration-dependent growth inhibition effect of catechin against both strains [63]. Actually, a peculiar bimodal behavior as a function of concentration was described for this flavonoid: at low concentrations, catechin exerted a beneficial effect on bacterial growth, but when the compound amount rose above a threshold value, a significant growth inhibition was observed on both strains tested [66]. This suggested the possible implication of catechin in the regulation of reactive oxygen species (ROS) production/elimination equilibrium. This hypothesis was confirmed through the analysis of ROS produced by oxidative damage caused by catechin, which suggested a time and concentration-dependent effect of the flavonoid. Similarly, Sinsinwar et al. reported a boost in ROS generation in two ATCC strains and five clinical isolates of MRSA, MSSA *S. aureus* upon catechin treatment. Moreover, a significant decrease in superoxide dismutase (SOD) and catalase (CAT) activity was found after treatment of those *S. aureus* strains with catechin at its MIC. Some polyphenols have been shown to inhibit bacterial growth by directly causing a change of membrane fluidity followed by membrane physical rupture, cytoplasm leakage, and outflow of some intracellular components [67]; therefore, also cell membrane integrity and permeabilization after catechin treatments was studied. For this purpose, the fluorescent dye propidium iodide was used, allowing to demonstrate the occurrence of an outer membrane damage after catechin incubation in all treated strains. Moreover, authors studied the changes in cell morphology by SEM analysis of bacterial liposomes models. These analysis highlighted catechin having a more deleterious effect on *B. subtilis* than *E. coli*, which was possibly because of the differences in the two bacterial cell walls’ composition. The presence of peptidoglycan in Gram-positive bacteria could implement the binding and absorption of catechin, whereas the LPSs on Gram-negative outer membranes may exert repulsive charge effect with the partially dissociated phenolic groups of the flavonoid, leading to a non-efficient binding of catechin to the membrane surface. An SEM image also showed membrane damage in *S. aureus* strains: the untreated cells were spherical and with smooth surfaces, whereas catechin-treated ones were deformed and inhomogeneous. Anyway, in both cases, SEM images of cell membranes after catechin treatment clearly showed membrane damage.

George et al. and He et al. screened the activity of other hydroxylated flavonoids: leucodelphinidin, quercetin, dihydroquercetin, and kaempferol activity against *E. coli* and *S. aureus* was investigated [68], whereas biochanin A, hesperetin, kaempferol, and (+)- catechin hydrate were tested only toward *E. coli* [69]. In the first study, the incubation of *E. coli* and *S. aureus* cells with a flavonoids-containing extract from *Penicillum setosum* led to the appearance of surface craters and holes up to complete cell rupture in both bacteria. When the extract was added to those bacteria, a leakage of Na^+^ and K^+^ ions to extracellular suspensions was observed, indicating cell membrane damage with concentration ranges in line with the MIC values retrieved. However, these membrane structure–perturbation effects were not just related to interaction with lipids. Therefore, a molecular docking-based screening was carried out toward different putative enzyme targets involved in the synthesis of type II fatty acids (β-ketoacyl ACP reductase, β-hydroxyacyl ACP dehydratase, and trans 2- enoyl ACP reductase) [70] and peptidoglycan (d-alanine:d-alanine ligase and penicillin binding protein-2) [71]. These enzymes are implicated in the fatty acid biosynthetic pathway, and their inhibition at various extents was described for polyphenolic molecules. Among the investigated compounds, leucodelphinidin showed the higher affinity to the putative targets, followed by dihydroquercetin, even though also quercetin was previously reported for its ability to inhibit d-alanine:d-alanine ligase enzyme [72]. These results further confirmed the higher efficiency of hydroxylated flavonoids in protein binding, even though the presence of hydrophobic moieties could enhance unspecific membrane binding effects. He et al. carried out similar studies on other flavonoids, using *E. coli* as a model bacterium. MIC values of tested compounds retrieved the following antibacterial efficacies: kaempferol > hesperetin > (+)-catechin hydrate > biochanin A. The same activity pattern was mirrored in membrane damage extents, TEM analysis showed maximum plasmolysis with kaempferol and minor damage with biochanin A [69]. Notably, when the same analysis was carried out on *M. tuberculosis*, measured MIC values showed an almost opposite trend, and the methoxylated biochanin A was the most effective compound. As the mycobacteria cell wall is the most hydrophobic bacteria envelope, this evidence suggested some selectivity of the different flavonoid for specific cell envelopes [73]. In addition, a study aimed at evaluating how flavonoids alter the fluidity of membranes, enhancing its rigidity and leading to membrane damage, was performed using a liposomal model membrane. Phosphatidylethanolamines (PEs), phosphatidylglycerols (PGs), and dipalmitoylphosphatidylethanolamine (DPPE) were assayed against an artificial *E. coli* Gram-negative membrane, using Raman spectroscopy to monitor flavonoid penetration into the lipid bilayer. Based on the results obtained, the antimicrobial mechanism of these flavonoids was suggested to be driven by their interaction with the hydrophilic region of cell membrane phospholipids.

Finally, some papers reported the effect of membrane disruption by different prenylated flavonoids. Pang et al. studied the antibacterial effects of morusin, which is a flavonoid carrying isopenthyl groups and extracted from *Morus alba* L. (Moraceae) on *S. aureus* and *Salmonella enterica* strains [74]. The lower MIC value was measured for the Gram-positive *S. aureus*, where damage to the cell membrane structure following the treatment was observed by TEM. The activity of morusin in *S. aureus* was related to its ability to impair the phosphatidic acid biosynthesis pathway and phospholipid-repair system. Indeed, the expression levels of different subunit of the fatty-acid-synthase (fabD, fabF, fabG, and fabH) were significantly downregulated in treated samples, indicating that morusin modulates the expression of genes involved in the phosphatidic acid biosynthesis pathway of *S. aureus*. Accordingly, the straight-chain fatty acids (SCFAs) and unsaturated fatty acids (UFAs) abundance was lowered, and the proportion of short-chain fatty acids was higher [74]. Moreover, Araya-Cloutier et al. published an extensive study in which 30 prenylated iso-flavonoids were tested against *Listeria monocytogenes* and *E. coli* and the results obtained underwent a Structure–antibacterial Activity Relationships (SAR) analysis [75]. First, the authors demonstrated an antibacterial activity for all the 30 tested flavonoids on *L. monocytogenes* whereas concentrations above 50 µg/mL were necessary for *E. coli* growth inhibition, thus suggesting that the prenyl group confers to polyphenols a much higher efficiency toward Gram-positive bacteria. However, the activity on EP inhibition seemed to give similar positive results, especially for monoprenylated isofavans and isofavones. Unexpectedly, these were also the best membrane permeabilizers of *E. coli* when compared to much more hydrophobic prenylated compounds.

A comparison between the result obtained by Sinsinwar et al. [64] and Araya-Cloutier et al. [75] suggested that both hydrophilic (catechin) and highly hydrophobic (highly prenylated) flavonoids have a lower efficiency as membrane permeabilizers and membrane damaging agents in Gram-negative bacteria than in Gram-positive ones. The hypothesis was advanced that the different permeabilization route followed by flavonoids in these microorganisms is the key parameter influencing their efficiency: in Gram-positive bacteria, the internalization of the bioactive compounds is mainly driven by hydrophobic interactions with the peptidoglycan. Conversely, in Gram-negative bacteria, the entrance mechanism of flavonoids involves their interaction with the negatively charged phospholipidic bilayer and a number of unspecific porins, the penetration through which is favored by amphipathic or moderately lipophilic features [75]. QSAR analysis correlating the experimental results to a bioinformatic model, to describe the effect of different structural features on flavonoids activity, were also carried out. Resulting data highlighted that prenylation at C8 provided a higher antibacterial activity for flavanones and at C6 for isoflavones. The presence of three (against Gram-positives) and four (against Gram-negatives) hydroxyl groups also seemed to potentiate antibacterial activity. To conclude, a model was constructed of an “optimal” antibacterial flavonoid structure, bearing one hydrophobic moiety, two aromatic rings, corresponding to the A- and B-ring of (iso)flavonoids, and one (for *L. monocytogenes*) or two (for *E. coli*) hydrogen bond acceptor projection moieties.

### 2.3. Flavonoids in Bioenergetic Status Perturbations

Flavonoids’ capability to disrupt the cellular bioenergetic status is mainly related to their role in ATP synthase inactivation. ATP synthase is the enzymatic complex responsible for ATP production through oxidative phosphorylation. Most studies regarding flavonoids and polyphenols exerted ATP synthase inactivation concerns the F1Fo ATP synthase from *E. coli*. It contains eight different subunits, some of which present in multiple copies, and the correct stoichiometry of the polyprotein is α_3_β_3_γδεab_2_c: F1 corresponds to α_3_β_3_γδε and contains three main catalytic sites, Fo is composed by ab_2_c, and it is responsible for proton transport [76]. The two portions of the complex can be foreseen as a biological rotary motor: proton gradient-driven clockwise rotation of subunit γ leads to ATP synthesis. Different molecules and antibiotics are reported to be ATP synthase inhibitors: 4-chloro-nitrobenzo-2-oxa-1,3-diazole (NBD-Cl), sodium azide (NaN_3_), aluminium fluoride (AlFx), and several naturally occurring antibiotics such as oligomycin, efrapeptins, aurovertins, and leucinostatins. Bedaquiline, an inhibitor of *M. tuberculosis* ATP synthase, has been approved by the United States Food and Drug Administration [77], and a few polyphenols such as resveratrol, piceatannol, quercetin, morin, and epicatechin were reported to inhibit *E. coli* ATP synthase [78,79]. The binding of these molecules to the protein complex occurs at the so-called polyphenol binding pocket, and it blocks the ATP synthase clockwise or anti-clockwise rotation of the γ-subunit [80]. Chinnam et al. reported a comprehensive study of *E. coli* ATP synthase inhibitory activity of seventeen bioflavonoid/polyphenol compounds never tested before for this activity [81]. Both purified F1-ATPase and membrane-bound F1Fo ATP synthase were used. This study showed that dietary bioflavonoids bind to *E. coli* ATP synthase and reversibly inhibit the enzyme. Based on the results of ATP synthase activity assays, the 17 investigated bioflavonoids were divided into three groups: potent inhibitors (≈0% residual activity), including (in decreasing order of potency) morin, silymarin, baicalein, silibinin, rimantadine, amantadine, and (−)-epicatechin; partial inhibitors (≈40–60% residual activity), including hesperidin, chrysin, kaempferol, diosmin, apigenin, genistein, and rutin; and weak inhibitors (≈80–100% residual activity), including genistein, galangin, luteolin, and daidzein.

These results were further confirmed by assaying the effect of these flavonoids on the growth of *E. coli* strain cultured in limiting glucose medium; the trend of the potency in inhibiting the growth of bacteria measured for the different compounds analyzed was similar to what was observed for ATPase inhibition. Based on these results, the authors performed some structural–activity relationship analyses. It was evident that the presence of sugar moieties bound to the flavonoid resulted in a significant reduction of their activity, which was probably due to the sterical hindrance in the polyphenol-binding pocket. Moreover, the presence of more hydroxy and methoxy groups on adjacent carbons, as in baicalein, epicatechin, silymarin, or silibinin, led to a higher inhibition of ATPase activity. In addition, the higher effect of quercetin compared to apigenin could be related to the two additional hydroxyl groups on its flavone skeleton. Interestingly, the varied extent of activities reported in this study for different flavonoids was not observed in the mitochondrial ATP synthase inhibition assays, which were previously described for similar molecules [82]. These results, along with the effect of growth inhibition on *E. coli* cells, suggested ATP synthase as a putative molecular target driving flavonoids antibiotic activity.

## 3. Enzymes Inhibitory Activity of Flavonoids in Bacteria: Direct and Indirect Mechanisms

The landscape of enzyme inhibitory activities of flavonoids, finally resulting in an antibacterial effect, is wide and varied. Generally, a targets pattern overview could be summarized in two main groups: (1) interactions with key enzymes involved in different anabolic and catabolic pathways, such as fatty acid and sterols synthesis, cell wall cross-linking construction, Krebs cycle, and glucose metabolism; and (2) inactivation of enzymes directly involved in pathogenic and antibiotic-resistance mechanisms, such as toxins and virulence factors transporters and antibiotic-inactivating enzymes.

In the work of Lee et al., the authors described an antimicrobial flavonoid, YKAF01 (3,6-dihydroxyflavone), which exhibits antibacterial activity against *E. coli* bacteria through the inhibition of β-ketoacyl acyl carrier protein (ACP) synthases [83]. ACP-synthase III (KAS III, also called acetoacetyl-ACP synthase), encoded by the *fabH* gene, is thought to catalyze the first elongation reaction of type II fatty acid synthesis in bacteria; KAS I (FabB), KAS II (FabF), and KAS III (FabH) belong to the FAS system and are predominant targets for the design of novel antibiotics [84,85]. In this study, the authors successfully identified YKAF01 flavonoid as a dual antimicrobial inhibitor of KAS I and KAS III, performed molecular docking studies to define the binding sites of YKAF01 on the two KASs, and evaluated the binding affinity between the compound and the two proteins using a fluorescence quenching experiment. Similar results were obtained by Xiaolan Xu, who evaluated the antifungal activity against *Penicillium notatum* of the ethanolic extracts of propolis from China and United States, which were both composed mainly by flavonoids (70% of the dry weight) [86]. The authors demonstrated that in the presence of propolis, the activities of succinate dehydrogenase (SDH) and malate dehydrogenase (MDH), which are related to energy metabolism processes including the tricarboxylic acid cycle and oxidative phosphorylation, were significantly decreased [87]. Moreover, these authors proved the interference of propolis on the synthetic pathways of sterol, especially on that leading to biosynthesis of ergosterol, which resides on fungal cell membranes and acts to maintain cell membrane integrity [88]. Another mechanism of action associated with the antibacterial effect of flavonoids involves the direct binding of virulence-associated proteins with the host cell [89]. Schneewind and colleagues brought definitive evidence that Sortase A (SrtA), an enzyme involved in the covalent linkage of some surface proteins of *S. aureus* to peptidoglycan, plays a key role in the display of surface proteins and in the virulence of this important human pathogen [90]. Ikhoon Oh et al. outlined that SrtA inhibitors might be promising candidates for the treatment and/or prevention of Gram-positive bacterial infections. Moreover, they showed that flavonoids extracted from *Sophora flavescens* Aiton (Leguminosae) were able to bind this enzyme and block its catalytic action. They identified the flavonoids present in the extract and showed that the prenylated flavonoid kurarinol exerts the strongest antibacterial activity [91].

A different way through which flavones are supposed to carry out their direct antibacterial action was proposed by Xu et al. These authors focused on the study of the Shuanghuanglian formula (SF), which is a renowned antimicrobial and antiviral traditional Chinese medicine composed by a mixture of *Lonicera japonica* Thunb. (Caprifoliaceae), *Scutellaria baicalensis* Georgi (Lamiaceae), and *Forsythia suspensa* Thunb. (Oleaceae) [92]. According to the Chinese Pharmacopoeia, SF is commonly administered in the forms of oral liquid, tablets, and injection, and it has long been used to treat acute respiratory tract infections, especially lung infection. Here, the authors demonstrated that the diglycosylated flavonoid lonicerin inhibited *P. aeruginosa* infections by blocking the function of the enzyme AlgE, which is an outer membrane protein involved in the translocation of alginate. During colonization of the lung, *P. aeruginosa* converts to a mucoid phenotype, which is characterized by the overproduction of the exopolysaccharide alginate. The secretion of newly synthesized alginate across the outer membrane is believed to occur through the outer membrane protein AlgE [93]. Lonicerin was found to be bound at the active site of the alginate secretion protein and was the first described potential AlgE antagonist derived from herbal sources.

As previously outlined, the combinations of antibiotics and flavonoids showed a synergistic activity, which is able to revert a resistance mechanism sometimes very efficiently. Eumkeb et al. showed that the use of a combination of flavonoids and antibiotic drugs was a potential strategy to prolong the effective life of antibiotics in the face of increasing antibiotic resistance [94]. Specifically, the flavonoids from smaller galanga, *Alpinia officinarum* Hance (Zingiberaceae), were used against β-lactam-resistant *S. aureus*, alone and in combination with β-lactam antibiotics, providing promising results. The observed antimicrobial activity was imputable to two different phenomena: the change in cell wall integrity, which significantly impaired protein synthesis, and the natural compounds direct inhibitory effect on the penicillin-binding protein. In this framework, Ramirez and Tolmasky focused their study on aminoglycoside antibiotics (AGs: gentamycin, amikacin, neomycin, streptomycin, spectinomycin, and tobramycin), showing that some flavonoids inhibited aminoglycoside-modifying enzymes (AMEs), acetyltransferases (AACs), nucleotidyltranferases (ANTs), or phosphotransferases (APHs) enzymes [95].

## 4. Virulence Factors: Flavonoid-Mediated Inhibition Mechanism

Virulence factors are molecules produced by bacteria to promote invasion, attack against eukaryotic cells, and evade host defenses, and they are the genesis of disease. They can be categorized into the following: (1) adherence factors, such as pili and fimbriae, which help bacteria to colonize mucosal sites; (2) invasion factors, promoting the invasion and translocation across the physiological barrier; (3) capsules, which can facilitate the opsonization of pathogens, (4) endotoxins, such as the components of Gram-negative outer membrane lipopolysaccharides, and (5) exotoxins, which are proteins that can act through various mechanisms. The search for natural agents that are able to inhibit or limit the action of virulence factors is a useful strategy to limit the spread of diseases due to bacterial infections and to face antibiotic resistance. Lee et al. evaluated the effect of 12 different flavonoids on the production of staphyloxanthin and α-hemolysin in *S. aureus*. Staphyloxanthin is a golden carotenoid pigment acting as a virulence factor enabling the detoxification of host-immune system-generated ROS, such as oxygen radical and hydrogen peroxide. Alpha-haemolysin is an exotoxin responsible for hemolysis of the host red blood cells. Among tested compounds, flavone (50 µg/mL) reduced the staphyloxantin production tenfold compared with the non-treated control group. A significant reduction of pigment levels was also detected after treatment with luteolin (25 µg/mL). Moreover, a real-time qRT-PCR experiment demonstrated that flavone (50 µg/mL) significantly inhibits the expression of the gene encoding for α-hemolysin (Hla) [96]. Recently, Stojkovìc et al. observed a downregulation of expression of staphyloxantin in *S. aureus* after treatment with methanolic extract of *Phlomis fruticosa* L. (Lamiaceae) at sub-MIC concentration [97]. *L. monocytogenes* is a common intracellular pathogen that usually grows in conditions used for food conservation, and it is the etiological agent of listeriosis, which is responsible for gastroenteritis, meningitis, and abortions. During the invasion phase, the bacterium produces listeriolysin O (LLO), which is a cholesterol-dependent cytolysin (CDC) that allows bacteria to escape from the phagosome and enter the cytosol of the host, disrupting the vacuole membrane. Moreover, the interaction between the toxin and cholesterol promotes structural changes in water-soluble monomer of LLO, which leads to pore formation in host cells membrane. LLO, such as other members of the pore-forming CDC family, is composed of four structural domains (D1-4), the first three of which are involved in toxin oligomerization and membrane disruption, while the fourth contributes to membrane binding. The toxin is produced as a water-soluble dimer or monomer, and the binding with membrane phospholipid bilayer is allowed by three hydrophobic loops (L1–L3) located in the carboxy terminal domain (D4). In particular, the interaction between cholesterol and the toxin occurs through contacts between the molecule and L1 and L2 of LLO. Based on the crucial role of this toxin in the bacterial hemolytic activity and genesis of *Listeria* infection, several researchers focused on the identification of LLO inhibitors. Wang et al. demonstrated that fisetin, a natural dietary flavonoid, inhibits the escape of toxin from phagosomes, facilitating its elimination from macrophages. Authors predicted the binding mechanisms of fisetin to LLO, identifying an interaction between the L2 and L3 toxin region and the compound, performing the molecular docking analysis, as reported in Figure 4. In addition, the authors observed that the toxin binding to fisetin induces a conformational change in LLO, making L1 and L2 sites poorly accessible to cholesterol, thus preventing the interaction of the toxin with membrane and limiting the hemolytic action of LLO [98]. Subsequently, they performed molecular docking studies and identified other flavonoids (myricetin, morin, chrysin, naringenin, and baicalein) that are able to bind the toxin more efficiently. Their results allowed confirming that the C1-C2 double bond of the flavonoids is one of the key moieties in the inhibitors of listerolysin O; in fact, naringenin, lacking this double bond, does not form the π–π interactions with Tyr427 of toxin [99]. A confirmation of these findings came a few years later from Nwabor et al., who observed the inhibitory effect of the flavonoids-rich *Eucalyptus camaldulensis* Dehnh (Myrtaceae) ethanolic extract on listeriolysin O production [100].

A completely different mechanism of action was demonstrated by Rajeshwari et al. studying the effects of cranberry juice against the periodontal pathogen *Porphyromonas gingivalis*. They mainly focused on two exotoxins: fimbrial proteins, which play a crucial role in the bacterial adhesion to the tissue surfaces, representing a necessary step for biofilm formation and tissue invasion, and gingipains, members of the trypsin-like cysteine proteinases family that cause vascular permeability enhancements and facilitate bacterial colonization. Cranberries, the ripe berries of *V. macrocarpon*, are a natural source of polyphenols, especially anthocyanins. The results obtained in this research showed that the incubation of *P. gingivalis* with a thermoreversible gel of cranberry juice concentrate induced a significant reduction in the expression of fimbrillin and (*fimA*) Lys-gingipain (*kgp*) genes, as detected by real-time qRT-PCR. Moreover, authors suggested that the cranberry juice concentrate prevented the multiplication of *P. gingivalis* by limiting the availability of the amino acids and peptides required for bacterial growth and inhibiting bacterial proteinases performing surrounding tissues destruction [101].

## 5. Flavonoids Involved in Antibiofilm Activities

Biofilms are well-organized communities of microorganisms involved in a self-produced extracellular matrix. Bacteria can develop biofilm as an alternative survival strategy for their growth and development, protecting the embedded cells from antimicrobial agents and host immune systems [102].

The biofilm formation consists of three steps: surface attachment, maturation, and dispersal. Firstly, bacteria produce EPS as extracellular polysaccharides, proteins, and other components involved in cell-to-surface adhesion and cell-to-cell cohesion during maturation. Subsequently, the dispersal of biofilm-embedded cells occurs via self-produced EPS-disrupting factors and other unknown mechanisms. So, dispersed cells can move into host organisms or in the environment [102].

In general, the mechanism underpinning biofilm formation in *S. aureus* is quite complex and involves environmental factors, quorum sensing, protease, DNase, and other regulators [103]. A pivotal role in biofilm formation is played by Quorum sensing (QS), which is a bacterial cell–cell communication process that involves production, detection, and response to extracellular signaling molecules called autoinducers (AIs). These molecules accumulate in the environment when the bacterial population density increases, modulating the expression of several bacterial genes. Gram-positive and Gram-negative bacteria use different types of QS systems. Gram-positive bacteria use peptides (AIPs), whereas Gram-negative bacteria use small molecules as acyl-homoserine lactones (AHLs) or other compounds, whose production depends on S-adenosylmethionine (SAM) [104]. The QS system in Gram-negative bacteria consists of three components: a LuxI synthase homolog, acyl-homoserinelactone (AHL) signaling molecules, and a LuxR receptor homolog [105]. In Gram-positive bacteria instead, peptide signaling molecules are carried out of the cell and are able to bind a membrane-associated two-component receptor. The receptor binding activates a signal transduction system, leading to the transcription of QS-regulated genes. In addition, for both types of bacteria, the QS system can also use furanosyl borate diester as a signaling molecule (named AI-2), which is responsible for interspecies communication [105]. Initiation of the QS signaling relies on cell density: when bacterial cell density exceeds a specific threshold, AIs bind the corresponding receptor and activate the QS system [106].

Generally, antibiotics are used to control Gram-positive and Gram negative pathogenic microbial growth and survival. However, it has been observed that biofilm-producing strains develop antibiotic resistance very fast, thus requiring the use of new antibacterial and antibiofilm compounds. Therefore, biofilms have a great impact on public health, causing about 65–80% of microbial diseases [107]. In this context, it has been reported that plant-derived extracts or isolated metabolites often exhibit good antibacterial and antibiofilm properties against several microorganisms. Many studies showed that flavonoids can interfere with the bacterial communication system (quorum sensing) or directly inhibit biofilm assembly.

De Vincenti et al. focused on investigating the antibiofilm performance of the extracts obtained from three seagrasses, a group of underwater marine flowering plants, among which the most promising was that from *Enhalus acoroides* (L.f.) Royle (Hydrocharitaceae). To this purpose, they incubated a biofilm-forming strain of *E. coli* with sub-toxic concentrations of that extracts. The results they obtained revealed an anti-adhesion activity, cell to surface, of the leaf extract, as demonstrated by the significant increase in the number of dispersed cells observed after bacteria treatment. This extract is mainly characterized by the presence of flavones as apigenin and luteolin and three kaempferol derivatives; studies suggested that the flavonoids were able to modulate bacterial cell–cell communication by suppressing the activity of the AI-2 [108]. Das et al. investigated the antimicrobial and antibiofilm potentiality of the ethyl acetate fraction of *Parkia javanica* (Lam.) Merr. (Leguminosae) fruit extract (PJE) and of three purified flavones (baicalein, quercetin, and chrysin) against *P. aeruginosa*. The antimicrobial activity analysis highlighted that at sub-MIC concentration, the PJE extract induced a significant reduction of microbial biofilm formation and the pure compounds showed a moderate activity. Docking analysis for baicalein, quercetin, and chrysin against the biofilm activity related proteins of *P. aeruginosa* was performed, showing that the three flavones have some binding affinity for the proteins of las and pil operon. In particular, LasR showed the highest binding affinity for baicalein and crysin, while PilT had the highest binding affinity for quercetin [109].

Morioka et al. demonstrated that the molecular chaperone DnaK is involved in the biofilm formation, and the chemical inhibition of DnaK cellular functions can prevent biofilm development. DnaK is a bacterial heat shock protein, playing a critical role in protein folding and refolding. In particular, DnaK increases the expression of CsgA and CsgB, which are two structural proteins involved in the aggregation curli fibers—extracellular functional amyloids produced by many *Enterobacteriaceae* [110].

Based on this observation, the authors hypothesized that small molecules inhibitors of DnaK could be used to prevent the production of biofilm formation in *E. coli*. Therefore, they studied the antibiofilm properties of myricetin, which is a flavonol widely distributed in plants. Results indicate that this compound inhibited cellular functions of DnaK that contribute to curli-dependent biofilm formation. These data were also confirmed by Pruteanu et al.; they showed that luteolin, myricetin, morin, and quercetin reduced the extracellular matrix inhibiting assembly of amyloid curli fibers in *E. coli*, although their effect was species-specific. Specifically, flavonoids stimulated macro-colony and biofilm formation in *P. aeruginosa*, whereas naringenin and the chalcone phloretin inhibited biofilm growth in *B. subtilis* [111].

Finally, Santos et al. studied the antibiofilm and anti-QS activities of the phenolic extracts obtained from the seeds and pulp of *Syzygium cumini* (L.) Skeels (Myrtaceae) (jambolan), obtaining promising results. Based on these, molecular docking calculations were performed, indicating that several among the compounds of pulp and seed extracts of jambolan, particularly dihydroquercetin, could potentially bind CviR protein, thus interfering with the QS system [112].

## 6. DNA Metabolism: DNA Gyrase as Target of Antimicrobial Molecule

DNA gyrase is a topoisomerase II, an ATP-dependent enzyme that plays a crucial role in transcription, DNA duplication, and chromosomal segregation. Structurally, it is an A_2_B_2_ hetero-tetrameric enzyme composed of two subunits: GyrA and GyrB. The first domain of GyrA, also called the G segment, binds specific DNA regions, while the second is a C-terminal domain; also, GyrB consists of the two domains, the first carrying the ATPase activity and the second being a topoisomerase-primase (TOPRIM) domain [113]. The knowledge of the structure of DNA gyrase makes it an attractive target for antibacterial molecules. In this context, Plaper et al. reported that quercetin inhibited the supercoiling activity of bacterial gyrase. The mechanism of action concerned the binding of the molecule to a 24 kDa fragment of *E. coli* GyrB, resulting in the inhibition of its ATPase activity [72]. In silico analyses, conducted in *M. tuberculosis* and *M. smegmatis*, also reveal the TOPRIM domain as a putative target of quercetin. This result was further confirmed by the experimental MIC values measured for that flavonoid against both bacteria [114]. Along the same research area, Gradisar et al. demonstrated a similar action mechanism for other flavonoids, such as catechins. Specifically, carrying out NMR experiments and molecular docking approaches, they showed that epigallocatechin gallate (EGCG), epicatechin gallate (ECG), and epigallocatechin (EGC) bind to the 24 kDa N-terminal fragment of GyrB. Among these molecules, the structure–activity relationship analyses showed a higher affinity of EGCG and ECG for the protein compared to what was observed for EGC. Therefore, it was assumed that the observed difference depended on the ability of gallate to interact efficiently with the active site region of the protein [115].

## 7. Conclusions and Future Perspectives

The functional and structural characterization of plants secondary metabolites has always been a prolific field in the pharmaceutical research, as a number of drugs and therapeutical agents were developed over time by modifying and optimizing natural bioactive molecules. Antibiotics are one of the main examples of this research trend. In latest years, the need for new classes of antibacterial substances has become an increasingly important issue to respond to pathogens resistance mechanisms, which are increasing at an alarming rate. In this framework, a growing deal of attention has been focused on flavonoids for their potential application as antibacterial agents and their multiple health benefits [116,117,118].

In many reported cases, significant antimicrobial activity was demonstrated for plant extracts, whose predominant constituents often consisted of flavonoids. However, in most cases, a complete analysis of the extracts composition was missing or, when it was present, the specific molecule(s) responsible for the observed antimicrobial effect, and its/their cellular targets were generally not described.

Therefore, the need is emerging to improve the analytical techniques and approaches used to identify individual flavonoids and map their specific interaction, to shed light on their specific mechanisms of action. The elucidation of the target proteins of the various flavonoids would have many advantages such as, for example, allowing to better clarify the synergistic interactions between these metabolites and drugs. Furthermore, defining in depth the interactome of fully structurally characterized flavonoids could pave the way for the design and synthesis of optimized bioactive molecules with a better activity profile. However, different factors hamper the advances in the biological studies on flavonoids. Examples encompass a great structure heterogeneity, a wide range of molecular interactions, and a scarce bioavailability and absorption, which further limits the investigation of flavonoids interactome at an in vivo level [119]. Future research aimed at improving the use of flavonoids as therapeutics or chemoprotective agents will face two main challenges: identifying their interaction targets and optimizing their bioavailability.

In this framework, the use of proteomic-based approaches to achieve flavonoids target identification appears to be very promising. Indeed, carrying out expression proteomic studies on biological samples treated or not with a bioactive flavonoid could allow defining the pathways involved in the biological action of these plant metabolites [120]. In addition, compound-centric proteomic experiments could lead to the identification of the specific protein these bioactive compounds interact with [121]. Emerging approaches, such as Drug Affinity Responsive Target Stability (DARTS) coupled with mass spectrometry, are giving optimal results as quick and easy approaches to identify potential protein targets for small molecules [122].

As mentioned above, alongside the definition of the mechanism of action, it is necessary to significantly improve the bioavailability of flavonoids after their administration. In fact, these compounds are absorbed inefficiently both orally and transdermally. Therefore, a great deal of effort is required from researchers to try to bridge the gap between the excellent bioactivities observed in vitro for several flavonoids and the quite low efficacy recorded in vivo for the same compounds. A technological solution allowing to overcome this important limitation would boost the use of flavonoids as novel antimicrobial therapeutic agents but also trigger a simple and effective trend in the biotechnological valorisation of raw plant material and natural wastes. A significant improvement in flavonoids bioavailability and bio-efficacy could be achieved using nano-delivery systems, such as nanospheres, nanocapsules, micro- and nanoemulsions, micelles, solid lipid nanoparticles, and nanostructured lipid capsules [123]. Over the past five years, there has been a sharp increase in articles describing the “phyto-fabrication” of nanoparticles based on copper, silver, gold, titanium, zinc, cadmium, gadolinium, or selenium-containing flavonoids [124]. These have been assembled from raw plant extracts but have also been used for the encapsulation of purified flavonoids. Several authors have shown that thanks to this innovative delivery system, it was possible to obtain an improvement in the beneficial qualities of flavonoids and have shown their possible applications to enhance the antioxidant and antimicrobial qualities of these compounds [125].

In conclusion, flavonoid derivatives offer an almost infinite and relatively still uncharacterized reservoir of therapeutical agents. According to the World Health Organization (WHO), medicinal plants should be considered as the best source to obtain a variety of drugs; however, only 1% of the approximately 500,000 plant species has been investigated from a phytochemical point of view so far [116,126]. Even lower is the number of molecular targets of these phytoproducts that have been identified. Therefore, the combined research for novel plant secondary metabolites and the implement of growing -omics techniques could still greatly expand the knowledge of flavonoids bioactivities, thus helping to best exploit what nature had already spontaneously evolved as a defense mechanism in response to infections and environmental stress.

## Figures and Tables

**Figure 1 pharmaceutics-13-00660-f001:**
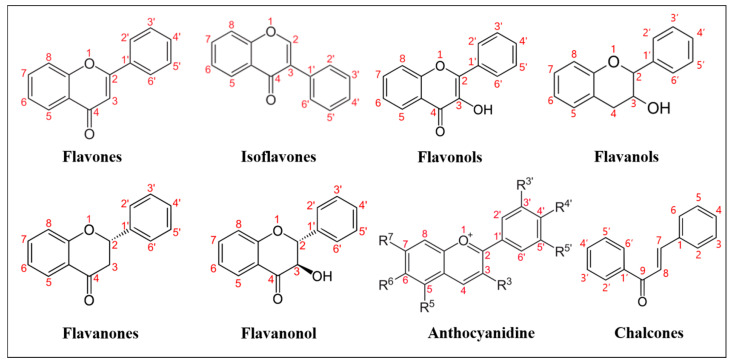
Basic structures of the main flavonoid classes.

**Figure 2 pharmaceutics-13-00660-f002:**
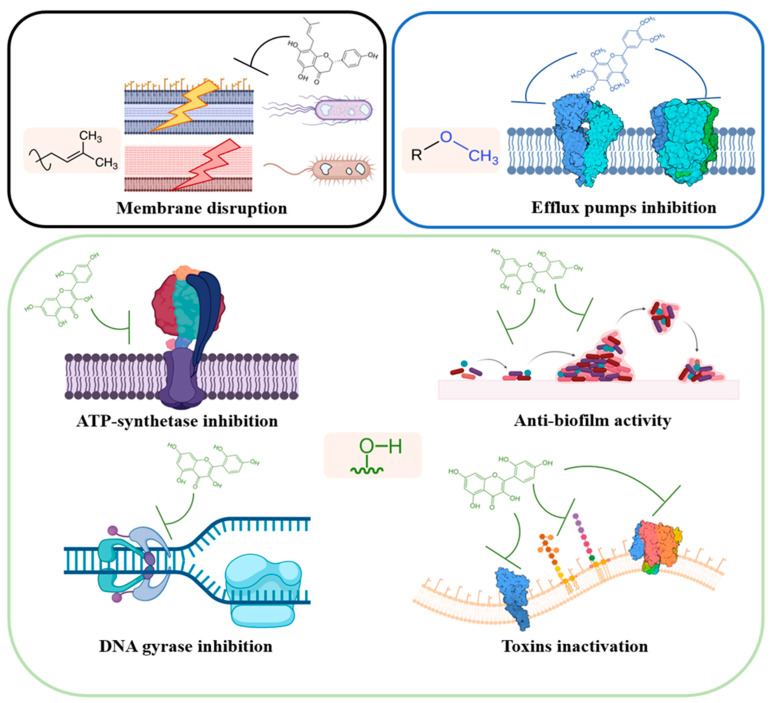
Main action mechanisms of flavonoids antibacterial activity. Flavonoids preferred substituents identified for each activity are also shown.

**Figure 3 pharmaceutics-13-00660-f003:**
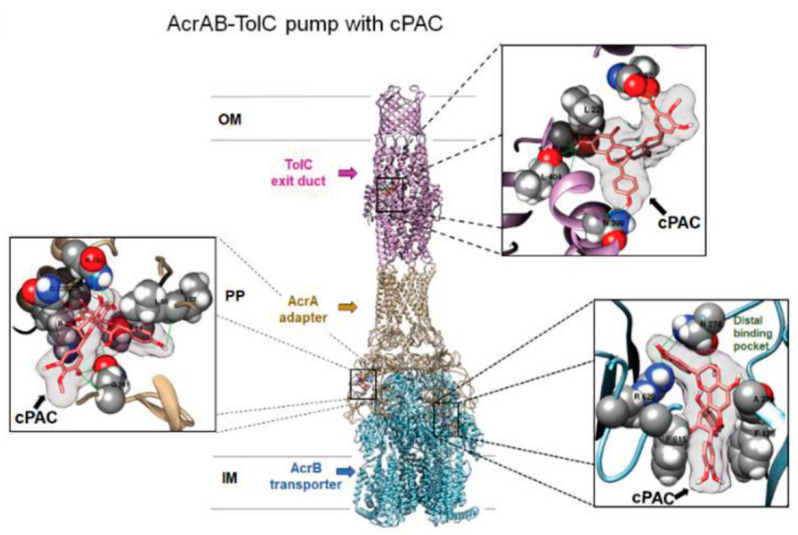
Molecular docking analysis of cPAC in the AcrAB–TolC efflux pump. (A) Complete side view with ribbon representation of docked complexes of efflux pump proteins with A-type cPAC molecule. The inset views show the electron density map (2F0–Fc) of cPAC in binding sites of multidrug efflux pump exit duct (TolC in pink), adapter (AcrA in gold), and transporter (AcrB in blue) proteins. OM, outer membrane; PP, periplasmic space; IM, inner membrane. Adapted from [40], Adv. Sci. 2019.

**Figure 4 pharmaceutics-13-00660-f004:**
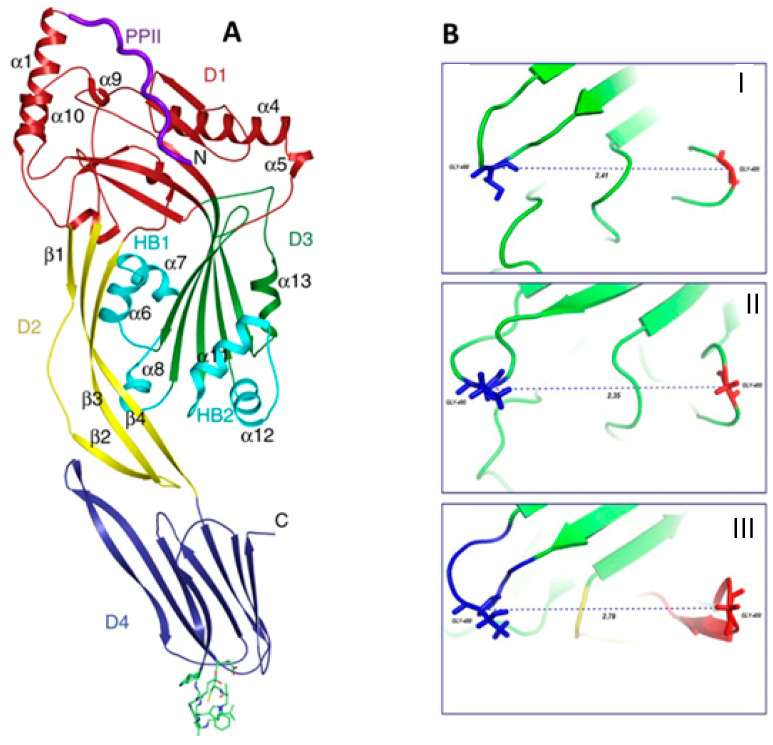
Molecular modeling of binding mechanisms of cholesterol (CHO) to listeriolysin O (LLO) and the effects of fisetin (FSN) on such binding. (**A**) LLO is organized into four domains (D1-D4). (**B**) I–III, Comparison of the distances (in nm) between the Cα atoms of Gly490 and Gly400 is shown in the structures of free LLO (I), the LLO–CHO complex (II), and the LLO–FSN complex (III). The average distance of the LLO–CHO complex is 2.35 nm. In the absence of ligand, the average distance between the defined points is 2.41 nm, and in the LLO–FSN complex, the distance is 2.79 nm. Adapted from [99], J. Infect. Dis. 2015.

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
