# Peer review of "Interactions with Microbial Proteins Driving the Antibacterial Activity of Flavonoids"

_pharmaceutics, 2021, doi:10.3390/pharmaceutics13050660_

Round 1

Reviewer 1 Report

In their manuscript entitled "Interactions with microbial proteins driving the antibacterial activity of flavonoids" Donadio et al. present a thorough review of the literature on studies performed on the interactions of bacterial proteins with flavonoids. Flavonoids are polyphenolic compounds produced by plants that can be generally characterized by a three-cyclic structure, with a wide range of strucutural variations due to the occurence of different substituents. These group of molecules have attracted a large interest due to their antimicrobial properties, either when used alone or in combination with antibiotics. Their natural origin is also an attractive point, as they can be extracted from plants without the need for chemical synthesis. However, as the authors highlight, many studies carried out so far are limited to the antimicrobial effects of extracts and molecular mechanisms underlying the antimicrobial activity of flavonoids reamins scarce. The manuscript presents a thorough review of the already known mechanisms of action of flavonoids and is therefore an important contribution to the advance of the kowledge in this research field. The manuscript is well organized and written, and there are only a few minor typos that should be corrected:

line 235: Compositae

line 265: "...a case a ..."

line 286: " nobiletin"

line 366: "hose" those?

line 423: S.

line 510: belong

line 524 and 527 SrtA and Sortase. If both refer to the same protein, it is better to use only one name.

line 593: Listeria

line 646: communication

line 649: bacterial

line 691:  CsgA and CsgB

Author Response

We thank the reviewer very much for his advice. We have made the corrections you indicated in the new versiono of the manuscript and we hope it is now suitable for publication.

Reviewer 2 Report

The manuscript presented for review entitled "Interaction with microbial proteins driving the antibacterial activity of flavonoids" is a very interesting full study of the topic.

As a reviewer, I only request minor corrections.

In key words should be added. Flavonoids and antibacterial activity.

Explain the concepts of MDR and DR on line 128-129.

Line 212 improved on pneumoniae.

The names of genes should be written well in italics - line 237, 508, 627,

line 646 correct to communication.

After applying corrections, the work can be sent for printing. 

Author Response

(The authors gave the same response as above.)

Reviewer 3 Report

The paper is interesting, however, I did not manage to understand if the images from the molecular modelling are yours or not. Maybe this aspect should be specified more clearly in the manuscript.

Author Response

We thank the reviewer very much for his advice. We provided more details concerning the figures  as you suggested in the new version of the manuscript and we hope it is now suitable for publication.